# Once-Weekly Insulin Icodec in Diabetes Mellitus: A Systematic Review and Meta-Analysis of Randomized Clinical Trials (ONWARDS Clinical Program)

**DOI:** 10.3390/biomedicines12081852

**Published:** 2024-08-14

**Authors:** Giuseppe Lisco, Anna De Tullio, Vincenzo De Geronimo, Vito Angelo Giagulli, Edoardo Guastamacchia, Giuseppina Piazzolla, Olga Eugenia Disoteo, Vincenzo Triggiani

**Affiliations:** 1Interdisciplinary Department of Medicine, School of Medicine, University of Bari “Aldo Moro”, Piazza Giulio Cesare 11, 70124 Bari, Italy; giuseppe.lisco@uniba.it (G.L.); annadetullio16@gmail.com (A.D.T.); vitogiagulli58@gmail.com (V.A.G.); edoardo.guastamacchia@uniba.it (E.G.); giuseppina.piazzolla@uniba.it (G.P.); vincenzo.triggiani@uniba.it (V.T.); 2Unit of Endocrinology, Policlinico Morgagni CCD, 95125 Catania, Italy; vdg@iol.it; 3Unit of Endocrinology, Diabetology, Dietetics and Clinical Nutrition, Sant Anna Hospital, San Fermo della Battaglia, 22020 Como, Italy

**Keywords:** Icodec, once-weekly basal insulin, type 1 diabetes, type 2 diabetes, ONWARDS, randomized clinical trials, meta-analysis

## Abstract

Background. One hundred years have passed since the discovery of insulin, which is one of the most relevant events of the 20th century. This period resulted in extraordinary progress in the development of novel molecules to improve glucose control, simplify the insulin regimen, and ameliorate the quality of life. In late March 2024, the first once-weekly basal analog Icodec was approved for diabetes mellitus, generating high expectations. Our aim was to systematically review and meta-analyze the efficacy and safety of Icodec compared to once-daily insulin analogs in type 1 (T1D) and type 2 diabetes (T2D). Methods. PubMed/MEDLINE, Cochrane Library, and ClinicalTrials.gov were searched for randomized clinical trials (RCTs). Studies were included for the synthesis according to the following prespecified inclusion criteria: uncontrolled T1D or T2D, age ≥ 18 years, insulin Icodec vs. active comparators (Degludec U100, Glargine U100, Glargine U300, and Detemir), phase 3, multicenter, double-blind or open-label RCTs, and a study duration ≥ 24 weeks. Results. The systematic review included 4347 patients with T1D and T2D inadequately controlled (2172 randomized to Icodec vs. 2175 randomized to once-daily basal analogs). Icodec, compared to once-daily basal analogs, slightly reduced the levels of glycated hemoglobin (HbA1c) with an estimated treatment difference (ETD) of −0.14% [95%CI −0.25; −0.03], *p* = 0.01, and I^2^ 68%. Patients randomized to Icodec compared to those on once-daily basal analogs had a greater probability to achieve HbA1c < 7% without clinically relevant or severe hypoglycemic events in 12 weeks from randomization with an estimated risk ratio (ERR) of 1.17, [95%CI 1.01, 1.36], *p* = 0.03, and I^2^ 66%. We did not find a difference in fasting glucose levels, time in range, and time above range between Icodec and comparators. Icodec, compared to once-daily basal analogs, resulted in a slight but statistically significant weight gain of 0.62 kg [95%CI 0.25; 0.99], *p* = 0.001, and I^2^ 25%. The frequency of hypoglycemic events (ERR 1.16 [95%CI 0.95; 1.41]), adverse events (ERR 1.04 [95%CI 1.00; 1.08]), injection-site reactions (ERR 1.08 [95%CI 0.62; 1.90]), and the discontinuation of treatments were similar between the two groups. Icodec was found to work better when used in a basal-only than basal-bolus regimen with an ETD in HbA1c of −0.22%, a probability of achieving glucose control of +33%, a probability of achieving glucose control without clinically relevant or severe hypoglycemia of +28%, more time spent in target (+4.55%) and less time spent in hyperglycemia (−5.14%). The risk of clinically relevant or severe hypoglycemic events was significantly higher when background glinides and sulfonylureas were added to basal analogs (ERR 1.42 [95%CI 1.05; 1.93]). Conclusion. Insulin Icodec is substantially non-inferior to once-daily insulin analogs in T2D, either insulin-naïve or insulin-treated. However, Icodec works slightly better than competitors when used in a basal-only rather than basal-bolus regimen. Weight gain and hypoglycemic risk are substantially low but not negligible. Patients’ education, adequate lifestyle and pharmacological interventions, and appropriate therapy adjustments are essential to minimize risks. This systematic review is registered as PROSPERO CRD42024568680.

## 1. Background

The discovery of insulin represents one of the most valuable scientific events of the 20th century, as it significantly contributed to the comprehension of diabetes mellitus pathophysiology and had relevant fallouts from a therapeutic viewpoint [1].

After the first identification of pancreatic islets by Paul Langerhans in 1869, more than 50 years passed until insulin was isolated for the first time by Sir Frederick Banting and Charles Best under the direction of John James Richard MacLeod at Toronto University (1921) [2]. Leonard Thompson was the first patient with type 1 diabetes (T1D) to receive the first insulin dose to control glucose levels, marking an extremely important event that would dramatically change the prognosis of future patients with insulinopenic diabetes. Banting and MacLeod’s discovery was then honored with the Nobel Prize in Physiology or Medicine in 1923.

Subsequent decades were characterized by the fervid development of the pharmaceutical industry and biotechnologies with the aim of (1) expanding the production and distribution of insulin worldwide, given the progressively growing demand for the hormone in North America and Europe; (2) improving insulin safety and tolerability; and (3) developing novel insulin analogs with a long half-life to overcome the need for multiple administrations of short-acting insulin [3].

Frederick Sanger in 1958 and Dorothy Hodgkin in 1969 isolated the primary sequence and quaternary structure of human insulin, respectively [4,5], opening the gate to the future development of synthetic and completely humanized insulin analog, which took place in 1978 by David Goeddel with recombinant DNA technology (and amplification in *Escherichia coli*) [6].

Over the last 45 years, we have observed significant progress in the fields of Diabetology, Biotechnology, and Pharmacology after the development and approval of several insulin analogs with a rapid, ultra-rapid, and ultra-slow length of action and pre-filled pens that simplified the handling of insulin regimens significantly, especially for patients treated with multiple daily injections (MDI). At the same time, we observed a growing contribution of technology with glucometers, insulin pumps, glucose sensors, bolus calculators, and integrated systems, allowing increasingly more comfortable and tailored insulin delivery and better glucose control (Figure 1).

A recent investigation focused on the development of modern agents with different routes of administration (e.g., oral, transdermal, or inhalation), extremely long half-life (i.e., once weekly insulins), and analogs conjugated with glucose sensors and conveyed by specific platforms (smart insulins), which can act in a glucose-dependent manner [7,8,9].

Nowadays, the prevalence of diabetes is around 10% of the adult population worldwide, with estimations indicating that 635 million people will have established diabetes by 2030 and 783 million (+46%) by 2045 [10]. Type 2 diabetes (T2D) is the most common cause of diabetes, representing 90% of all cases, while T1D is less common (8–10%). Trends in insulin prescriptions in diabetes indicate a relevant increase in the number of insulin users among T2D patients, with around 17.4% and 52% of them, respectively, on basal-only and basal-bolus regimens [11]. Despite the novel agents currently available for T2D, the percentage of patients who are candidates for insulin treatment is expected to increase over the following decades due to the extension of life expectancy and the absolute increase in the number of patients living with T2D. As another issue, only 1 in 4 insulin users with T2D achieve their glucose targets, making frequent therapy adjustments compulsory, including a switch to other basal analogs, to improve glucose control [12]. The lack of adherence to insulin regimens, especially for patients on MDI, with the suboptimal frequency of glucose checks and therapeutic inertia of prompt insulin titration, represent the most common causes of insulin failure [13]. Overcoming these barriers can result in better glucose management in patients with diabetes.

## 2. Progress in Once-Weekly Insulins

Adequate adherence to pharmacological treatment contributes significantly to achieving and maintaining tailored glucose control, as guidelines recommend, especially in patients with MDI. Novel administration strategies have been studied and proposed, such as once-weekly administered drugs. Once weekly administered drugs represent a significant innovation in the pharmacological management of T2D, as demonstrated mainly by incretin-based injective treatments, such as glucagon-like peptide 1 receptor agonists (GLP-1RAs) and dual GLP-1 and glucose-dependent insulinotropic peptide receptor agonists [14,15,16,17]. Possible drawbacks of once-weekly insulins, such as insulin Icodec, are related to their prolonged half-life, leading to less frequent insulin dose adjustments compared to once-daily analogs that, in turn, could generate some concerns in managing glucose variability, especially in insulinopenic diabetes while using the basal-bolus regimen. Therefore, it is mandatory to address the efficacy and safety of insulin Icodec in various insulin regimens, particularly the basal-bolus regimen, in both types of diabetes, which was the aim of our systematic review. Moreover, insulin Icodec direct costs are expected to be higher than those of currently available basal insulin analogs. Large-scale, cost-effective trials are needed to comprehensively estimate the direct and indirect costs of insulin Icodec compared to once-daily insulin analogs.

Ultra-long compared to once-daily insulin analogs should not simply have a longer half-life; instead, they should ensure the lower variability of plasma concentration, a stronger affinity for serum albumin, and significantly lower affinity for insulin receptors, slowing down the receptor-mediated clearance of the insulin analog.

The first ultra-long insulin analog, Icodec (Awiqli^®^), was approved in March 2024 to treat T2D in Europe [18]. The novel analog should be administered subcutaneously once a week in sites conventionally used to administer other insulin analogs, including the thigh, abdomen, and upper arm [19]. Icodec can be marketed in three different packages, including pens of 1 mL, 1.5 mL, and 3 mL with a standard concentration of 700 IU/mL [20]. The Icodec dose should be calculated and adjusted weekly. The starting dose in insulin-naïve patients should be 70 IU/week, a dose paralleled to 10 IU per day of once-daily insulins. Patients who are switched from other basal insulins to Icodec should be replaced by maintaining a 1:1 ratio with the weekly dose of the basal analog, except for the first administration, which is recommended to increase the dose by 50 to 100% according to baseline-fasting glucose levels. The steady state is achieved after four weeks, and titration should be accomplished weekly (±20 IU) with a recommended fasting/pre-breakfast self-monitored glucose target of 80–130 mg/dL [21].

The pharmacokinetic profile of Icodec is not affected by mild, moderate, or severe hepatic impairment. A slight but statistically significant increase in Icodec exposure was reported along with a declining glomerular filtration rate; nevertheless, the clinical relevance of this phenomenon could be negligible and easily managed with proper titration [22].

Icodec results from molecular bioengineering that introduce several changes in the native structure of other insulin analogs starting from an oral insulin prototype (OI388) [23]. The addition of a C20 fatty diacid-containing side chain (acylation) induces robust and reversible binding to serum albumin, while three amino acid substitutions (chain A, 14E; chain B, 16H, and 25H) provide molecular stability and reduce binding to insulin receptor and clearance [24,25]. Overall, these changes prolong the half-life of Icodec, resulting in 196 h, which is compatible with once-weekly dosing. Moreover, the lower affinity of insulin Icodec, compared to native insulin, for both the insulin and insulin-like growth factor-1 receptors were proven to reduce the mitogenic effect of Icodec in various human cells.

Another once-weekly insulin analog is currently under investigation, namely, the Basal Insulin Fc (BIF, LY3209590, or insulin Efsitora alfa), which is composed of a novel single-chain variant of insulin fused to a human immunoglobulin G2 fragment and crystallizable region of an antibody domain using a peptide linker [26]. Three phase 2 trials have already been completed, and the published results demonstrate that insulin Efsitora alfa is effective and safe in patients with T1D [27] when compared to once-daily insulin Degludec U100 and both T2D insulin-naïve [28] and insulin users [29]. Five phase 3 trials are ongoing to investigate the efficacy and safety of insulin Efsitora alfa as the most relevant part of the once-weekly Insulin Therapy clinical program [30].

## 3. Efficacy and Safety of Insulin Icodec: The State of the Art

### 3.1. Phase 2 Trials

A summary of the phase 2 trial results is shown in Table 1. Icodec, compared to Glargine U100, was administered daily and achieved a non-inferiority endpoint that improved glucose control with a similar risk of level 2 and 3 hypoglycemia in insulin-naïve T2D individuals [31].

Titration is essential to achieve optimal glucose control without increasing the risk of hypoglycemia, as demonstrated by another phase 2 trial [32]. The best result in terms of improving glucose control and lowering the risk of hypoglycemia was obtained when Icodec, compared to Glargine U100, was titrated at ±28 IU/week to maintain a fasting/prebreakfast self-monitored glucose level between 80 and 130 mg/dL. The switch to Icodec vs. Glargine U100 in T2D patients who failed to achieve adequate glucose control with other basal insulins was slightly better in terms of the glucose control attained in 15 weeks, especially when a loading dose of Icodec (+100%) was administered at the switching time [33].

### 3.2. Phase 3 Trials

#### 3.2.1. Overview of the ONWARDS Clinical Program

The efficacy and safety of insulin Icodec were extensively assessed over six randomized clinical trials (RCTs) of the clinical program ONWARDS (Table 2) [34,35,36,37,38,39]. Five trials were conducted on T2D patients: 3 in insulin-naïve (ONWARDS 1, 3, and 5) and 2 in insulin-treated (ONWARDS 2 and 4) patients. Only one trial was conducted on T1D (ONWARDS 6). Icodec was compared to once-daily basal insulin types, namely Glargine U100 (ONWARDS 1, 4, and 5), Glargine U300 (ONWARDS 5), and Degludec U100 (ONWARDS 2, 3, and 5), to assess both the efficacy and safety of the novel once-weekly analog.

Each trial included a 2-week pretrial screening and a 5-week post-trial safety follow-up in which the patients were followed after treatment discontinuation for residual adverse events. The trials ONWARDS 1, 5, and 6 also included a 26-week extension phase during which patients continued the trial treatments in the same way as from the randomization to detect and register efficacy and safety endpoints until the study’s completion.

The primary endpoint of all trials was to compare the mean absolute changes in glycated hemoglobin (HbA1c) levels from baseline to study completion, which was set at 26 weeks in ONWARDS and 2, 3, 4, and 6 and 52 weeks in the remaining cohorts. Secondary endpoints included fasting glucose control, time in range (TIR), hypoglycemic risk, other safety outcomes, and patient satisfaction [40].

#### 3.2.2. Procedures

##### Pretrial Antihyperglycemic Treatment

Any antihyperglycemic drug was allowed before the study entry and during the trials at the same pretrial dose. Sulfonylureas or glinides were discontinued (ONWARDS 1, 2, and 4) or the pretrial dose halved (ONWARDS 3 and 5) because of unacceptable gain in the risk of hypoglycemia when combined with basal insulins.

In ONWARDS 4 and 6, all participants were switched from any pretrial prandial analog to insulin Aspart.

##### Starting Dose and Titration of Basal Analogs

The starting dose of Icodec and once-daily basal analogs was 70 IU per week (10 IU/day) in insulin-naïve individuals (ONWARDS 1, 3, and 5). In insulin-treated patients (ONWARDS 2, 4, and 6), the starting dose of both Icodec and once-daily basal analogs was the same as the weekly dose of the pretrial basal analog. For the first dose of Icodec, an additional 50% one-time dose was administered. In ONWARDS 6, T1D individuals with a baseline HbA1c > 8% received a 100% once-time additional dose of Icodec in addition to the first administration only.

Basal analogs were titrated weekly using a treat-to-target approach to achieve fasting/prebreakfast glucose levels of 80–130 mg/dL. In ONWARDS 5, the titration of Icodec was assisted by a dose guidance system integrated with a dose recommendation algorithm. Therefore, patients randomized to Icodec received specific training to run the system. Investigators carried out the titration of Degludec at their personal discretion as per standard practice.

##### Continuous Glucose Monitoring

Continuous glucose monitoring (CGM) was allowed in ONWARDS 1, 2, and 4. It was double-masked over the last four weeks of trials, with only a statistical aim to collect and analyze additional metrics of glucose control, including TIR (70–180 mg/dL), time above range, or TAR (>180 mg/dL), and time spent in clinically relevant hypoglycemia (<54 mg/dL).

In ONWARDS 6, patients with T1D were educated to wear and use an open CGM system to monitor glucose values during the study. However, as per the protocol, any CGM-based insulin adjustment was prohibited.

The GCM system used in all trials was Dexcom G6^®^.

##### Safety Endpoints

Safety endpoints were observed and reported over the entire study, including the extension phase and 5-week follow-up. Safety endpoints included any adverse events, such as serious and severe adverse events, events probably and possibly related to insulin use, hypersensitivity, injection-site reactions, hypoglycemia (overall, combined clinically relevant and severe, and nocturnal), and weight gain.

Hypoglycemic events were classified according to a standardized three-level severity scale as follows: level 1 hypoglycemia for glucose levels ranging from 55 to 70 mg/dL, level 2 or clinically significant hypoglycemia for glucose levels ≤ 54 mg/dL, and level 3 or severe hypoglycemia to indicate an event characterized by altered mental and/or physical status requiring assistance for the treatment of hypoglycemia, regardless of glucose levels [41].

##### Satisfaction and Compliance Questionnaires

The Diabetes Treatment Satisfaction Questionnaire and Treatment-Related Impact Measure for Diabetes were used as specific tools to assess secondary endpoints on diabetes satisfaction and compliance in insulin-treated patients in ONWARDS 2, 5, and 6 only.

#### 3.2.3. Methods

##### Searching, Screening, and Selection of Studies

Two operators (G.L. and A.D.T.) searched databases and registries, including PubMed/MEDLINE, Cochrane Library, and ClinicalTrials.gov, from 1 November 2020 to 9 August 2024, for RCTs assessing the efficacy and safety of insulin Icodec from the ONWARDS clinical program. Keywords and Medical Subject Heading (MeSH) terms included the following: “icodec”, “once-weekly basal insulin*”, “once-weekly basal analogue*”, “basal insulin*”, “glargine u100”, “glargine u300”, and “degludec u100”.

Databases were searched independently by each operator to mitigate possible biases. The prespecified clinical question was as follows: “Is insulin Icodec more effective and safer than once-daily basal analogs in improving glycemic parameters of patients with diabetes mellitus who failed to achieve glucose control with non-insulin agents or previous insulin treatment?”.

Records were screened and selected by each operator and then compared. The flow diagram illustrating the process of identification, screening, and inclusion of RCTs for this systematic review is shown in the Appendix A.

The other three operators (O.E.D., E.G. and V.T.) checked the literature for possible external sources of RCTs.

##### Inclusion Criteria

Studies were selected based on the following inclusion criteria: patients with uncontrolled T1D or T2D, age ≥ 18 years, insulin Icodec vs. active comparators (Degludec U100, Glargine U100, Glargine U300, and Detemir) alone or in combination with prandial analogs, phase 3, multicenter, double-blind or open-label RCTs, and a study duration of 24 weeks or more.

##### Exclusion Criteria

Non-randomized observational studies and case series were excluded.

##### Extraction and Synthesis: Comprehensive Details of RCTs

Two operators (G.L. and A.D.T.) extracted data from RCTs. The details of each RCT were extensively reviewed and collected (Table 1). The details include information on the study population, study design and duration, inclusion criteria, baseline antihyperglycemic drugs and the management of pretrial drugs during the trials, frequency of comorbidities, intervention, comparators, sample size, number of patients who completed the “in-trial” period, primary, secondary and safety endpoints, main baseline characteristics, main post-trial characteristics, starting and final weekly doses of insulin analogs, details on basal insulin titration, and additional information.

##### Extraction and Synthesis: Efficacy Endpoints

Efficacy endpoints were selected according to their clinical relevance, study design, and the heterogeneity of data reporting across all trials. These included (a) the mean absolute change in HbA1c levels from baseline to study completion, summarized as estimated treatment difference (ETD) between the two study groups; (b) the probability of achieving acceptable glucose control (i.e., HbA1c < 7%), summarized as the estimated risk ratio (ERR) or a chance to obtain a specific outcome; (c) the probability of achieving acceptable glucose control without clinically relevant or severe hypoglycemia (i.e., HbA1c < 7% without level 2 or level 3 hypoglycemia, by combining efficacy with a safety endpoint), summarized as ERR or the chance to obtain a specific outcome; (d) the mean absolute difference in TIR after the study completion, summarized as the ERR between the two study groups; and (e) the mean absolute change in fasting plasma glucose (FPG) from baseline to study completion, summarized as ETD between the two study groups.

Technical remark: efficacy endpoints (b) and (c) should be intended as early efficacy endpoints since they are estimated in 12 weeks from randomization in line with clinical practice and current recommendation, suggesting that HbA1c levels should be checked after 2 to 3 months from any therapy adjustment in patients with inadequate glucose control.

##### Extraction and Synthesis: Safety Endpoints

Safety endpoints were selected according to their clinical relevance, study design, and the heterogeneity of data reporting. These include (a) the mean absolute difference in TAR after the study completion, summarized as ETD between the two study groups; (b) the mean absolute change in body weight from baseline to study completion, summarized as ETD between the two study groups; (c) the probability of presenting with level 2 or level 3 hypoglycemia (combined endpoint), summarized as ERR for the outcome to occur; (d) the probability of presenting with any adverse event, summarized as ERR for the outcome to occur; (e) the probability of presenting with any adverse event probably or possibly related to basal insulin, summarized as ERR for the outcome to occur; (f) the probability of presenting with serious adverse events, summarized as ERR for the outcome to occur; (g) the probability of presenting with serious adverse events probably or possibly related to basal insulin, summarized as ERR for the outcome to occur; and (h) the probability of presenting with injection-site reactions, summarized as ERR for the outcome to occur.

Technical remark: safety analyses were carried out considering the number of patients who experienced a specific event (one or more times) from the total number of participants included in the safety analysis set.

##### Intention-to-Treat Analysis

All statistics were calculated according to an intention-to-treat analysis. Statistical analyses were conducted on two different clusters of patients according to the prespecified endpoints.

Efficacy endpoints were analyzed using the full-analysis set (randomized participants) and data from the “in-trial” period (from randomization to the last contact, withdrawal, or death).

Safety endpoints were assessed using the safety analysis set (randomized participants who received at least one dose of study drugs) from the “on-treatment” period (from randomization to the trial end).

##### Participants Who Completed the Trials

The “in-trial” period was completed by more than 90% of randomized participants, except for the Icodec arm in ONWARDS 5 (completion rate 89.1%), with a similar discontinuation rate between the two study groups.

Technical remark: the intention-to-treat analysis aims to reduce the attrition bias due to the relevant dropouts of participants during the follow-up. However, compared to a per-protocol analysis, it was less informative on the real effect of treatments in each stage of the follow-up. A high and symmetric completion rate ensures the readability of intention-to-treat analyses.

##### Assessment of the Risk of Bias and Publication Bias

The risk of included studies was estimated with the RoB2 assessment tool for individual randomized, parallel-group trials [42]. All trials were extensively evaluated in 5 separate domains, exploring the randomization process, the deviation from the intended interventions, missing data, the measurement of outcomes, and the selection of reported results. The risk of bias was estimated for each outcome (efficacy and safety), and each domain was rated as low, moderate (some concerns), or high (relevant).

Technical remark: primary and secondary endpoints, as well as additional exploratory assessments, were prespecified, except for some evaluations that were carried out post hoc, so when the preliminary or advanced results of RCTs were already available.

With specific regard to the endpoints of interest for our meta-analysis, it should be mentioned that only two variables were listed as not prespecified analyses, namely the variable “probability of experiencing combined clinically significant (level 2) or severe (level 3) hypoglycemia with a given treatment” in ONWARDS 2, and the variable “number of hypoglycemic alerts” in ONWARDS 4 and 6.

Hence, we chose to evaluate the domain bias number five (the selection of the reported results) of ONWARDS 2, 4, and 6 with some concerns (Figure 2).

The existence of publication bias for the primary outcome was verified by a funnel plot (Appendix A).

##### Software for Statistics

Forest plots and sensitive analyses were performed by RevMan 5.4.1 with a random-effect model, considering a *p*-value < 0.05 as statistically significant. Heterogeneity was assessed by I^2^. The level of heterogeneity was considered substantial in the case of I^2^ > 60%, leading us to explore the possible causes of heterogeneity in the specific result by subgroup analyses [43].

We used the standard error, 95% confidence interval (CI), and interquartile range to estimate the standard deviation when missed.

#### 3.2.4. Results

##### Synthesis of Data from the ONWARDS Clinical Program

The systematic review and meta-analysis included, for the primary endpoint (the mean absolute change in HbA1c from baseline to study completion), 4347 patients with T1D and T2D inadequately controlled and randomized to receive Icodec (2172) or once-daily basal analogs (2175) for 26 consecutive weeks or more.

The baseline-weighted mean level of HbA1c from the ONWARDS clinical program was 8.4%. Icodec, compared to once-daily insulin analogs, improved glucose control with an ETD of −0.14% [95%CI −0.25; −0.03], *p* = 0.01, and I^2^ 68% (Figure 3).

Patients randomized to Icodec, compared to those on once-daily basal analogs, had a 16% higher chance to achieve optimal glucose control (i.e., HbA1c < 7%) over 12 weeks from randomization but this difference was not statistically significant with an ERR of 1.16 [95%CI 0.95; 1.42], and I^2^ 84% (Figure 4).

Patients randomized to Icodec, compared to those randomized to once-daily basal analogs, had a statistically significant 17% greater chance to achieve optimal glucose control without clinically relevant or severe hypoglycemia (i.e., HbA1c < 7%, without level 2 or 3 hypos) over 12 weeks from randomization (ERR of 1.17 [95%CI 1.01; 1.36], *p* = 0.03, and I^2^ 66%) (Figure 5).

The baseline-weighted mean FPG from the ONWARDS clinical program was 174.7 mg/dL. ETD between Icodec and once-daily basal insulin doses was 2.44 mg/dL in favor of once-daily basal insulins, but this result was not statistically significant [95%CI −2.95; 7.82], I^2^ 70% (Appendix A).

Data from CGMs indicated that TIR was slightly higher with Icodec than once-daily basal insulins with an ETD of 1.64% (around 23 min/day), which was not statistically significant with 95%IC [−1.65; 4.93] and I^2^ 79%. At the same time, the ETD in TAR between patients on Icodec compared to those on once-daily basal insulins was −2.34% [95%CI −5.51; 0.83], and I^2^ 76%, parallel to a non-statistically significant lower exposure to potentially dangerous hyperglycemia (≥180 mg/dL) at around 35 min/day (Appendix A).

The baseline-weighted mean body weight from the ONWARDS clinical program was 85.8 kg. Treatment with basal insulin analogs increased body weight; however, Icodec, compared to once-daily insulins, induced a slight but statistically significant weight gain of 0.62 kg [95%CI 0.25; 0.99], *p* = 0.001, and I^2^ 25% (Figure 6).

The combined endpoint summarizing the risk of clinically relevant or severe hypoglycemic events (combined level 2 or level 3 hypoglycemia) revealed that a greater but not statistically significant number of patients randomized to Icodec than once-daily basal analogs had hypoglycemic events, with an ERR of 1.16 [95%CI 0.95; 1.41], I^2^ 75% (Figure 7).

Adverse events, mostly mild-to-moderate, were reported in a more relevant number of participants randomized to Icodec than in once-daily basal analogs with an ERR of 1.04 [95%CI 1.00; 1.08], *p* = 0.07 and I^2^ 0%, indicating an almost statistically significant 4% increase in the overall risk of adverse events. However, only a minority of events was finally adjudicated as possibly or probably related to basal insulin use, with a non-statistically significant difference in specific endpoint, with an ERR of 1.22 [95%CI 0.99; 1.49], *p* = 0.07, and I^2^ 20% (Appendix A).

Serious adverse events were registered in a lower absolute number of patients randomized to Icodec than those given once-daily basal insulin doses with an ERR of 0.94 [95%CI 0.78; 1.13] and I^2^ 0%, which was not statistically relevant. Serious adverse events possibly or probably related to basal insulin use were reported in an equal number of patients randomized to Icodec rather than those given once-daily basal insulin doses with a non-significant ERR of 1.08 [95%CI 0.66; 1.76] and I^2^ 13% (Appendix A).

The number of patients who presented with injection-site reactions was similar between the two arms, with an ERR of 1.08 [95%CI 0.62; 1.90] and I^2^ 21% (Appendix A).

##### Exploring the Heterogeneity of the Results

The meta-analysis revealed substantial heterogeneity in the results from the RCTs of the ONWARDS clinical, especially among the efficacy endpoints, TAR, and the risk of combined level 2 or 3 hypoglycemia. The leading causes of heterogeneity were explored across several factors, including baseline demographics, glucose control, insulin-naïve vs. insulin-treated patients, the type of once-daily basal insulin used as a comparator to Icodec, the concomitant use of sulfonylureas or glinides during the trials, the basal-bolus vs. basal only regimen, and the risk of bias (low vs. moderate risk).

Subgroup analyses revealed that heterogenicity disappeared when grouping the results of trials according to the type of intervention. More precisely, the efficacy and safety of Icodec were examined by considering how it worked in the context of a basal only rather than in a basal-bolus regimen. When used in the context of the basal-only regimen, Icodec obtained better results in terms of glucose control (ETD in HbA1c of −0.22%), probability to achieve glucose control (+33%) and glucose control without clinically relevant or severe hypoglycemia (+28%), with more time spent in target (+4.55%, around 65 min/day) and less time spent in hyperglycemia (−5.14%, 74 min/day) (Table 3).

The combined risk of experiencing level 2 or 3 hypoglycemia was statistically significant in ONWARDS 6 but not in ONWARDS 4. Patients were treated with basal-bolus regimens in both trials, but only patients with T1D had a statistically significant increase in the risk of combined level 2 or 3 hypoglycemia. Moreover, the risk of clinically relevant or severe hypoglycemic events was significantly higher when background glinides and sulfonylureas were included as an add-on to basal analogs (ONWARDS 3 and 5) with an ERR of 1.42 [95%CI 1.05; 1.93], and I^2^ 0%.

## 4. Discussion

The results of this systematic review and meta-analysis provide an update on Insulin Icodec in both T1D and T2D beyond previously published results [44,45,46,47]. So far, insulin Icodec has been demonstrated to be slightly better than daily administered basal analogs in terms of glucose control and the chance to achieve targeted glucose levels without a relevant gain in the risk of hypoglycemia in T2D. Specific comparisons between Icodec and either Glargine U100 or Degludec U100 showed that Icodec was slightly superior to Degludec U100 and equal to Glargine U100 in improving glucose control. At the same time, Icodec, compared to Degludec U100, increased the risk of any hypoglycemic events with moderate-to-substantial heterogenicity of the results [48].

Icodec was found to perform equally in insulin-naïve and insulin-treated T2D individuals, indicating that baseline antihyperglycemic treatment does not affect the clinical response to Icodec when stated de novo or switched from another basal analog [49].

Our data indicate that Icodec is effective in reducing HbA1c levels starting from 26 and up to 78 weeks of treatment. Compared to once-daily basal insulins, namely Glargine U100, Glargine U300, and Degludec U100, Icodec provide a statistically significant absolute mean change in HbA1c of −0.14%. Moreover, we found that Icodec, compared to once-daily basal insulins, increases the probability of achieving adequate glucose control (i.e., HbA1c < 7%) safely without level 2 or 3 hypoglycemic events by 17%. The therapeutic targets can be obtained after 12 weeks of treatment, which means just in time or soon before the second check of HbA1c levels after therapy adjustment.

Overall, the above-mentioned glycemic benefits did not translate into relevant changes in TIR registered by a CGM system during the last four weeks of the trials. No difference in FPG was also reported, probably because of the rigorous treat-to-target approach that investigators used in all trials. No data on FPG were assessed or reported in ONWARDS 5, which is a trial in which different methods of titration were applied to the two study groups. Patients randomized to insulin Icodec were guided by an app-based algorithmic tool to titrate basal insulin weekly, while patients randomized to once-daily basal insulins received instruction directly from investigators as per standard practice. It is unclear and complicated to predict how missing data could have influenced the cumulative weight of the FPG endpoint in our meta-analysis.

Most patients experienced slight weight gain during the trials, estimated at 2 kg as a mean. Icodec was responsible for an additional weight gain of 0.62 kg over once-daily basal analogs. This effect could be attributable to a slightly higher weekly dose of Icodec than basal analogs in all trials.

The number of patients experiencing hypoglycemic events was similar for Icodec and once-daily basal insulin doses in T2D [50] but not T1D (ONWARDS 6), where Icodec was associated with a statistically significant higher risk of clinically relevant or severe hypoglycemic events. This result cannot be explained entirely by the concomitant administration of a prandial analog in the context of a basal-bolus regimen since the risk of level 2 or 3 hypos was not increased in T2D (ONWARDS 4). The main explanation is the specific effect of Icodec on glucose variability in T1D and requires more investigation [51]. The combined risk of clinically relevant or severe hypoglycemic events was also higher among patients on Icodec than once-daily basal analogs in trials where sulfonylureas and glinides were not discontinued (ONWARDS 3 and 5). Although the pretrial dose of secretagogues was halved during both trials, the combination of sulfonylureas and glinides with Icodec resulted in a 42% increase in the risk of potentially dangerous hypoglycemic events. Therefore, this specific association should be avoided in clinical practice.

Safety endpoints were similar between Icodec and once-daily basal analogs. The overall risk of any adverse events was not statistically significant, and it was mitigated by the systematic revision of any signs and symptoms potentially related to insulin use (possibly or probably).

Injection-site reactions, as well as serious adverse events, were infrequent and statistically similar between the two groups.

Last, insulin Icodec works better when used in a basal regimen only rather than in the context of a basal-bolus regimen, resulting in a greater ETD in HbA1c (−0.22%), with a higher chance of achieving HbA1c < 7% (+33%) and HbA1c < 7% without clinically relevant or severe hypoglycemia (+28%), and a higher TIR (around 65 min/day) and lower TAR (−74 min/day) compared to once-daily basal analogs. These results were statistically relevant and occurred independently of pretrial treatments (both insulin and non-insulin agents) but could be attributable to a poorer baseline glucose control and lower duration of diabetes (ONWARDS 1, 2, 3, and 5).

## 5. Study Limitations

The leading limitation of the ONWARDS program trials is the sample size, which is adequate for the primary outcome (changes in HbA1c) only but does not allow subgroup analyses that are desirable for this kind of trials.

Second, most relevant safety outcomes, such as the estimation of hypoglycemic risk, were not included in the prespecified analyses and were calculated post hoc, thus potentially reducing the level of evidence of this relevant outcome.

Third, data from real-time glucose monitoring were scarce. CGM data were available only in four trials, but only in one (ONWARDS 6, in T1D) were data registered and analyzed during the entire study period. Consequently, information on glucose variability, daytime, and nocturnal hypoglycemic risk were limited and required to be better analyzed with additional trials.

## 6. Conclusions

Insulin Icodec, the first approved once-weekly insulin analog, provides evidence of substantial non-inferiority compared to once-daily basal analogs in diabetes management in both insulin-naïve and insulin-treated patients who failed to achieve adequate glucose control. Icodec works slightly better than once-daily basal analogs in T2D individuals, especially when used in the context of basal-only rather than a basal-bolus regimen.

Weight gain is highly predictable after insulin initiation, and Icodec confirms this well-known trend. Providing patients with effective non-pharmacological and pharmacological intervention is reasonable to avoid weight gain or promote weight loss when necessary.

Specific trials are expected to address the impact of Icodec on glycemic variability in individuals with T1D. Moreover, other trials are needed to comprehensively evaluate the best clinical scenario in which insulin Icodec can be cost-effective compared to once-daily insulin analogs.

## Figures and Tables

**Figure 1 biomedicines-12-01852-f001:**
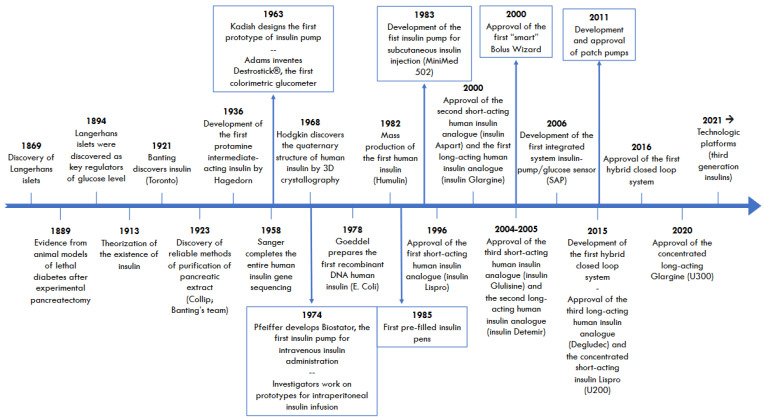
Timeline summarizing the most relevant discoveries and events in the fields of Diabetology, Biotechnology, and Pharmacology that have characterized the last 100 years.

**Figure 2 biomedicines-12-01852-f002:**
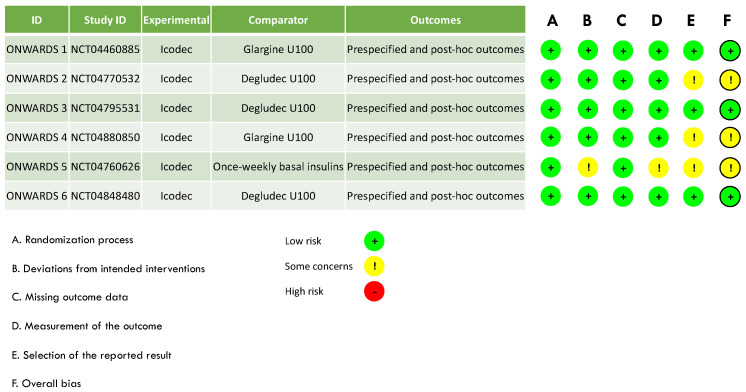
Risk of bias of RCTs included in the systematic review and meta-analysis from the ONWARDS clinical program.

**Figure 3 biomedicines-12-01852-f003:**
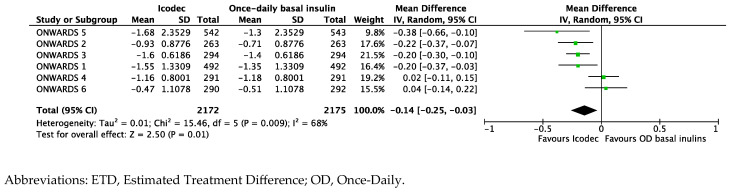
Forest plot of meta-analysis for mean change in Glycated Hemoglobin (ETD, %) from baseline to study completion (intention-to-treat analysis).

**Figure 4 biomedicines-12-01852-f004:**
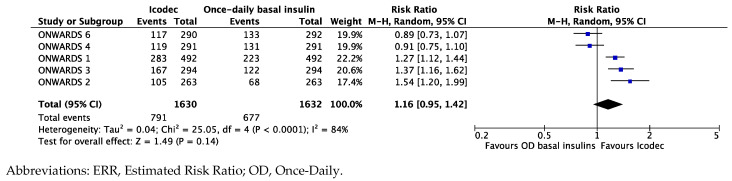
Forest plot of meta-analysis for probability (ERR) to achieve optimal glucose control (i.e., HbA1c < 7%) from baseline to 12 weeks (intention-to-treat analysis).

**Figure 5 biomedicines-12-01852-f005:**
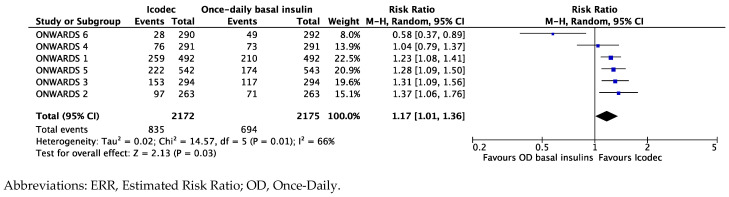
Forest plot of meta-analysis for probability (ERR) to achieve optimal glucose control (i.e., HbA1c < 7%) without clinically relevant (level 2) or severe (level 3) hypoglycemic events from baseline to 12 weeks (intention-to-treat analysis).

**Figure 6 biomedicines-12-01852-f006:**
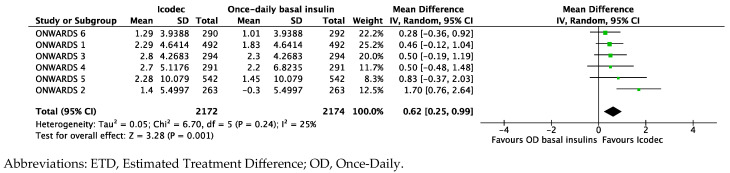
Forest plot of meta-analysis for mean change in body weight (ETD, kg) from baseline to study completion (intention-to-treat analysis).

**Figure 7 biomedicines-12-01852-f007:**
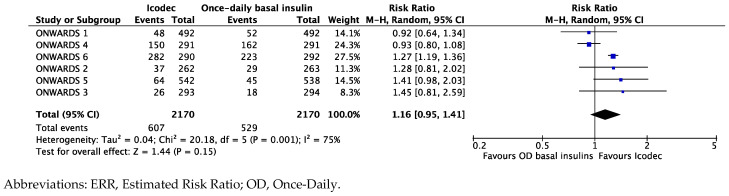
Forest plot of meta-analysis for probability (ERR, %) of experiencing clinically relevant (level 2) or severe (level 3) hypoglycemic events from baseline study completion (intention-to-treat analysis).

**Table 1 biomedicines-12-01852-t001:** Overview of phase 2 trials.

	NCT03751657 [27]	NCT03951805 [28]	NCT03922750 [29]
Study design	26-week double-blind, RCT	16-week open-label, randomized, treat-to-target, titration trial with glucose monitoring	16-week open-label, randomized, treat-to-target, switching trial with glucose monitoring
Population	T2D	T2D	T2D
Intervention	Once-weekly insulin Icodec	Once-weekly insulin Icodec Titration A (80–130 mg/dL = adjustment ± 21 IU/week); Titration B (80–130 mg/dL = ±28 IU/week); Titration C (70–108 mg/dL = ±28 IU/week)	Once-weekly insulin Icodec A) with loading dose (a 100% increase from the initial dose) B) without loading dose
Comparator	Once-daily insulin Glargine U100	Once-daily insulin Glargine U100 Titration (80–130 mg/dL = ±4 IU/day)	Once-daily insulin Glargine U100
Baseline characteristics	247 insulin-naïve participants, mean HbA1c 8%, metformin ± DPP-IV inhibitors	205 insulin-naïve participants, mean HbA1c 8.1%, any oral antihyperglycemic agents	154 insulin users (10–50 IU/day): Detemir, Degludec U100, Glargine U100, Glargine U300, mean HbA1c 7.9%
Main findings	Mean change from baseline in HbA1c: −1.33% Icodec vs. −1.15% Glargine U100 (*p* = 0.08) Hypoglycemia (levels 2 and 3): 0.53 events per patient-year Icodec vs. 0.46 events per patient-year Glargine U100 (RR 1.09; 95%CI, 0.45 to 2.65)	Mean change in TIR (baseline to 15–16 weeks) Icodec A: from 57.0% to 76.6% Icodec B: from 55.2% to 83% Icodec C: from 51.0% to 80.9% Glargine U100: from 55.3% to 75.9% Level 2 hypoglycemia (<54 mg/dL, events per patient-year of exposure) Icodec A: 0.05 Icodec B: 0.15 Icodec C: 0.38 Glargine U100: 0.00 No level 3 hypos were observed.	Mean change in TIR (baseline to 15–16 weeks) Icodec A: from 58.9% to 72.9% Icodec B: from 54.5% to 66.0% Glargine U100: from 58.7% to 65.0% Mean change in HbA1c Icodec A: from 7.9% to 7.1% Icodec B: from 7.9% to 7.4% Glargine U100: from 7.9% to 7.4% Level 1 and 2 hypos were similar (among the 3 groups), and no level 3 hypos were registered

Abbreviations: HbA1c, Glycated hemoglobin; IU, International Unit; T2D, Type 2 Diabetes.

**Table 2 biomedicines-12-01852-t002:** Comprehensive overview of the ONWARDS clinical program.

	ONWARDS 1 [30] (NCT04460885)	ONWARDS 2 [31] (NCT04770532)	ONWARDS 3 [32] (NCT04795531)	ONWARDS 4 [33] (NCT04880850)	ONWARDS 5 [34] (NCT04760626)	ONWARDS 6 [35] (NCT04848480)
Sponsored	Yes	Yes	Yes	Yes	Yes	Yes
Population	Insulin-naïve T2D	Basal insulin-treated T2D	Insulin-naïve T2D	Basal bolus-treated T2D	Insulin-naïve T2D	T1D
Inclusion criteria	Age ≥ 18 yrs, baseline HbA1c 7–11%, baseline BMI ≤ 40 kg/m^2^	Age ≥ 18 yrs, baseline HbA1c 7–10%	Age ≥ 18 yrs, baseline HbA1c 7–11%, baseline BMI ≤ 40 kg/m^2^	Age ≥ 18 yrs, baseline HbA1c 7–10%	Age ≥ 18 yrs, baseline HbA1c > 7%, for whom insulin treatment is required	HbA1c < 10% At least 1 year of basal-bolus regimen
Study design	Randomized, open-label, treat-to-target phase 3a trial	Randomized, open-label, active-controlled, multicentric, treat-to-target phase 3a trial	Randomized, double-masked, double-dummy, active-controlled, treat-to-target phase 3a trial	Randomized, open-label, multicentric, treat-to-target, non-inferiority trial	Randomized, open-label, multinational trial	Randomized, multicenter, open-label, active-controlled, parallel-group, treat-to-target, phase 3a trial
Study duration, weeks	78 (52 + 26 of extension safety phase) + a 5-week follow-up	26 + a 5-week follow-up	26 + a 5-week follow-up	26 + a 5-week follow-up	52 + a 5-week follow-up	52 (26 + 26 of extension safety phase) + a 5-week follow-up
Pretrial antihyperglycemic drugs	Any non-insulin drugs allowed	Once or twice-daily basal insulins ± non insulin antihyperglycemic agents	Any non-insulin drugs allowed	Any basal-bolus regimen ± non insulin antihyperglycemic agents (>90 days)	Any non-insulin drugs allowed	Basal-bolus regimen (any analogues allowed)
Handling of pretrial antihyperglycemic drugs at the randomization	Pretrial drugs confirmed, except secretagogues	Pretrial drugs confirmed, except secretagogues	Pretrial drugs confirmed at the same dose, including secretagogues (initial dose was reduced by 50%)	Pretrial drugs confirmed, except secretagogues	Pretrial drugs confirmed at the same dose, including secretagogues (initial dose was reduced by 50%)	Pretrial prandial insulins were switched to insulin Aspart
Comorbidities	NA	NA	Arterial hypertension (65%), hepatic steatosis (12.5%), coronary artery disease (10%), renal impairment (8.5%)	NA	Arterial hypertension (70%), hepatic steatosis (9.8%), coronary artery disease (8.5%)	NA
Intervention	Once-weekly insulin Icodec	Once-weekly insulin Icodec	Once-weekly insulin Icodec + once-daily placebo	Once-weekly insulin Icodec + insulin Aspart	Once-weekly insulin Icodec	Once-weekly insulin Icodec + insulin Aspart
Comparators	Once-daily insulin Glargine U100	Once-daily insulin Degludec U100	Once-daily insulin Degludec U100	Once-daily insulin Glargine U100 + insulin Aspart	Once-daily basal insulins (Glargine U100 or Glargine U300 or Degludec U100)	Once-daily insulin Degludec U100 + insulin Aspart
Sample size: n	Icodec: 492 Glargine: 492	Icodec: 263 Degludec: 263	Icodec: 294 Degludec: 294	Icodec: 292 Glargine: 291	Icodec: 542 OD Basal: 543	Icodec: 290 Degludec: 292
Completed the “in-trial” period: %	Icodec: 96.5% Glargine: 97.4%	Icodec: 97.7% Degludec: 96.2%	Icodec: 95.9% Degludec: 96.2%	Icodec: 94% Glargine: 92%	Icodec: 89.1% OD Basal: 90.8%	Icodec: 90% Glargine: 95%
Primary Outcome	Mean change from baseline to study completion in HbA1c	Mean change from baseline to study completion in HbA1c	Mean change from baseline to study completion in HbA1c	Mean change from baseline to study completion in HbA1c	Mean change from baseline to study completion in HbA1c	Mean change from baseline to study completion (week 26) in HbA1c
Secondary outcomes	Mean change from baseline to study completion in FPG, TIR, weekly insulin dose, body weight	Mean change from baseline to study completion in FPG, TIR, weekly insulin dose, body weight, diabetes satisfaction	Mean change from baseline to study completion in FPG, weekly insulin dose, body weight	Mean change from baseline to study completion in FPG, TIR, weekly insulin dose, body weight	Mean change from baseline to study completion in diabetes satisfaction and compliance, weekly insulin dose and body weight	Mean change from baseline to study completion in FPG, TIR, HbA1c (week 52), body weight, diabetes satisfaction
Safety outcomes	Adverse events, hypoglycemic episodes (levels 1, 2, and 3)	Adverse events, hypoglycemic episodes (levels 1, 2, and 3), daytime and nocturnal hypos	Adverse events, hypoglycemic episodes (levels 1, 2, and 3)	Adverse events, hypoglycemic episodes (levels 1, 2, and 3)	Adverse events, hypoglycemic episodes (levels 1, 2, and 3), daytime and nocturnal hypos	Adverse events, hypoglycemic episodes (levels 1, 2, and 3), daytime and nocturnal hypos
Age, yrs: mean ± sd	Icodec: 59.1 ± 10.1 Glargine: 58.9 ± 9.9	Icodec: 62.3 ± 9.8 Degludec: 62.6 ± 8.4	Icodec: 58 ± 10 Degludec: 59 ± 10	Icodec: 59.7 ± 10.1 Glargine: 59.9 ± 9.9	Icodec: 59.1 ± 10.8 OD Basal: 59.4 ± 10.2	Icodec: 44.1 ± 14.1 Degludec: 44.3 ± 14.1
Diabetes duration, yrs: mean ± sd	Icodec: 11.6 ± 6.7 Glargine: 11.5 ± 6.8	Icodec: 16.5 ± 8.4 Degludec: 16.9 ± 7.9	Icodec: 10.5 Degludec: 10.7	Icodec: 18 ± 9.1 Glargine: 16.3 ± 7.7	Icodec: 11.9 ± 6.9 OD Basal: 12 ± 7.6	Icodec: 20 ± 13.2 Degludec: 19 ± 12.9
Baseline HbA1c, %: mean ± sd	Icodec: 8.5 ± 1 Glargine: 8.4 ± 1	Icodec: 8.17 ± 0.77 Degludec: 8.1 ± 0.77	Icodec: 8.55 ± 1.11 Degludec: 8.48 ± 1.01	Icodec: 8.29 ± 0.86 Glargine: 8.31 ± 0.9	Icodec: 8.96 ± 1.6 OD Basal: 8.88 ± 1.5	Icodec: 7.59 ± 0.96 Degludec: 7.63 ± 0.93
Final HbA1c, %: mean ± sd	Icodec: 6.93 ± 1.33 Glargine: 7.12 ± 1.11	Icodec: 7.2 ± 0.81 Degludec: 7.42 ± 0.97	Icodec: 7 ± 1.09 Degludec: 7.2 ± 0.98	Icodec: 7.14 ± 0.85 Glargine: 7.12 ± 0.85	Icodec: 7.24 ± 2.01 OD Basal: 7.61 ± 2.7	Icodec: 7.15 ± 1.1 Degludec: 7.1 ± 1.1
Baseline FPG, mg/dL: mean ± sd	Icodec: 185.3 ± 49 Glargine: 185.7 ± 51.7	Icodec: 155.2 ± 47 Degludec: 150.7 ± 40.9	Icodec: 187 ± 54 Degludec: 176 ± 46	Icodec: 165.6 ± 54 Glargine: 172.8 ± 63	Icodec: NA OD Basal: NA	Icodec: 179 ± 74 Degludec: 172 ± 72
Final FPG, mg/dL: mean ± sd	Icodec: 125.2 ± 37 Glargine: 125.4 ± 37.3	Icodec: 129.1 ± 29.3 Degludec: 117.7 ± 26	Icodec: 127 ± NA Degludec: 127 ± NA	Icodec: 137 ± 41 Glargine: 132 ± 39	Icodec: NA OD Basal: NA	Icodec: 163.9 ± NA Degludec: 138.3 ± NA
Baseline BMI, kg/m^2^: mean ± sd	Icodec: 30 ± 4.8 Glargine: 30.1 ± 5.1	Icodec: 29.5 ± 5.2 Degludec: 29.2 ± 4.9	Icodec: 29.9 ± 5.2 Degludec: 29.2 ± 5.1	Icodec: 30.5 ± 5 Glargine: 30 ± 5	Icodec: 32.6 ± 7 OD Basal: 33 ± 6.9	Icodec: 26.8 ± 5 Degludec: 26.2 ± 4.5
Baseline body weight, kg: mean ± sd	Icodec: 85.2 ± 17.7 Glargine: 84.3 ± 17.6	Icodec: 83.7 ± 18.4 Degludec: 81.5 ± 17.1	Icodec: 85.8 ± 20.1 Degludec: 83.2 ± 18.2	Icodec: 85.5 ± 17.6 Glargine: 83.1 ± 17.3	Icodec: 93.2 ± 22.5 OD Basal: 94.3 ± 21.5	Icodec: 78.6 ± 17.6 Degludec: 77.1 ± 16.8
Final body weight, kg: mean ± sd	Icodec: 87.03 ± 0.21 Glargine: 86.57 ± 0.21	NA	Icodec: 87.3 ± NA Degludec: 86.8 ± NA	Icodec: 88.2 ± NA Glargine: 85.3 ± NA	Icodec: 96 OD Basal: 95.2	Icodec: 79.9 ± NA Degludec: 78.1 ± NA
Starting dose of basal insulin (IU/week)	Icodec: 70 Glargine: 70	1:1 ratio with pretrial basal insulins	Icodec: 70 Degludec: 70	1:1 ratio with pretrial basal insulins	Icodec: 70 Degludec: 70	1:1 ratio with pretrial basal insulins
Weekly insulin dose at the study completion: n (IU)	Icodec: 214 Glargine: 222	Icodec: 268 Degludec: 244	Icodec: 204 Degludec: 186	Icodec: 305 Glargine: 279	Icodec: 227 OD Basal: 185	Icodec: 132 Degludec: 161
Treat-to-target approach	Yes, 80–130 mg/dL	Yes, 80–130 mg/dL	Yes, 80–130 mg/dL	Yes, 80–130 mg/dL	NA	Yes, 80–130 mg/dL
Titration of basal insulin	Icodec: ±20 per week Glargine: ±3 per day	NA	Icodec: ±20 per week Degludec: ±3 per day	Icodec: ±20 per week Glargine: ±3 per day	Icodec: algorithmic-assisted titration Degludec: at the discretion of investigators	Icodec: ±20 per week Degludec: ±3 per day Aspart: dose adjustment (week 0 to 8) or carbohydrate-counting
Frequency of insulin dose adjustment	Once a week	Once a week	Once a week	Once a week	Once a week	Once a week
Additional metrics/tools	Yes, double-blind CGM (weeks 48–52)	Yes, double-blind CGM (weeks 22–26)	None	Yes, double-blind CGM (weeks 22–26)	ICOBOT engine for guiding Icodec titration only	Yes, open CGM (whole study, but not used for insulin titration)
Satisfaction questionnaire	None	Yes, DTSQ	None	None	Yes, DTSQ, TRIM-D	Yes, DTSQ

Abbreviations: CGM, Continuous Glucose Monitoring; DTSQ, Diabetes Treatment Satisfaction Questionnaire; FPG, Fasting Plasma Glucose; HbA1c, Glycated hemoglobin; IU, International Unit; NA, Not Assessed/reported; OD, Once a Day; T1D, Type 1 Diabetes; T2D, Type 2 Diabetes; TIR, Time in Range; TRIM-D, Treatment Related Impact Measure for Diabetes.

**Table 3 biomedicines-12-01852-t003:** Subgroup analyses exploring the heterogeneity of results in efficacy and safety from the ONWARDS clinical program.

Main Outcomes with Significant Heterogenicity (I^2^ >60%)	Icodec in the Context of Basal Regimen (Subgroup 1-ONWARDS 1, 2, 3, and 5)	Icodec in the Context of Basal-Bolus Regimen (Subgroup 2-ONWARDS 4 and 6)
ETD [95%CI], I^2^ in HbA1c	**−0.22% [−0.29; −0.14], I^2^ 0%** **Favors Icodec**	0.03% [−0.08; 0.13], I^2^ 0%
ERR [95%CI], I^2^ in probability to achieve HbA1c < 7% *	**1.33 [1.21; 1.47], I^2^ 2%** **Favors Icodec**	0.90 [0.79; 1.02], I^2^ 0%
ERR [95%CI], I^2^ in probability to achieve HbA1c < 7% without experiencing clinically relevant or severe hypoglycemia	**1.28 [1.17; 1.39], I^2^ 0%** **Favors Icodec**	0.79 [0.44; 1.42], I^2^ 81%
ETD [95%CI], I^2^ in TIR **	**4.55% [2.01; 7.08], I^2^ 0%** **Favors Icodec**	−0.88% [−2.97; 1.20], I^2^ 28%
ETD [95%CI], I^2^ in FPG ***	0.1 mg/dL [−3.01; 3.20], I^2^ 0%	8.05 mg/dL [−12.99; 29], I^2^ 91%
ETD [95%CI], I^2^ in TAR **	**−5.14% [−7.27; −3.01], I^2^ 0%** **Favors Icodec**	0.07% [−2; 2.15], I^2^ 0%
ERR [95%CI], I^2^ in the probability of experiencing combined level 2 or 3 hypoglycemic events ****	1.12 [0.98; 1.15], I^2^ 0%	1.09 [0.76; 1.57], I^2^ 95%

* Subgroup 1-ONWARDS 1, 2, and 3; Subgroup 2-ONWARDS 4 and 6. ** GCM data were available only from ONWARDS 1, 2, 4, and 6. Subgroup 1—ONWARDS 1 and 2; Subgroup 2—ONWARDS 4 and 6. *** FPG not assessed or reported in ONWARDS 5. Subgroup 1-ONWARDS 1, 2, and 3; Subgroup 2-ONWARDS 4 and 6. Substantial variability is attributable to the results of the ONWARDS 6 only. **** Substantial variability attributable to the results of the ONWARDS 6 only. Statistically significant differences are in bold. Abbreviations: Estimated Risk Ratio, ERR; Estimate Treatment Difference, ETD.

## Data Availability

Data supporting the findings of this systematic review and meta-analysis are available from the first, last, or corresponding author on reasonable request.

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
