# Peer review of "Once-Weekly Insulin Icodec in Diabetes Mellitus: A Systematic Review and Meta-Analysis of Randomized Clinical Trials (ONWARDS Clinical Program)"

_biomedicines, 2024, doi:10.3390/biomedicines12081852_

Round 1

Reviewer 1 Report

Comments and Suggestions for Authors

The authors need to revise the guidelines for the abstract. The structure of the abstract need to match the Journal requirement.  The introduction is properly written and it demonstrates insulin development and discovery timeline. 

Progress in Once-weekly insulin section: 
the authors cover thoroughly the benefits of once weekly insulin, it tends to emphasize the positive aspects without sufficiently addressing potential drawbacks. For example, the discussion on patients adherence improvement is extensive but it lack a critical examination (Does less frequent dosing could lead to complacency in self-monitoring or insulin administration adherence?). 
Another point can be raised and need to be added in the section is the limitations and potential areas where daily insulin may still have a favor specially in fine-tuning dosing. 
adjustments. 

Also, I have another concern regarding the economic aspects of once-weekly insulin compared with the once daily insulin. Is it cost-effective ? 

Methods section: 

1. Why did the authors choose 24 weeks as a minimum duration for the included RCT? 

2. It is recommended to add PRISMA figure to illustrate the process of selection and the numbers of articles found in each step. 

3. There is no mention about who performed data extraction. Was it independently by multiple reviewers? 

4. The authors did not mention how did they handle missing data. 

Discussion section is written well however, the conclusion should be restructured in cohesive manner.

Author Response

I thank the review for comments and suggestions to improve the paper's quality. Herein, you can find a point-by-point reply to comments and suggestions.

Question 1. The authors need to revise the guidelines for the abstract. The structure of the abstract need to match the Journal requirement.  The introduction is properly written and it demonstrates insulin development and discovery timeline. 

Reply 1. The abstract structure was adjusted following the journal guidelines. Please check the paper for details (changes are in red).

-----

Question 2. 

Progress in Once-weekly insulin section: 

the authors cover thoroughly the benefits of once weekly insulin, it tends to emphasize the positive aspects without sufficiently addressing potential drawbacks. For example, the discussion on patients adherence improvement is extensive but it lack a critical examination (Does less frequent dosing could lead to complacency in self-monitoring or insulin administration adherence?). 

Another point can be raised and need to be added in the section is the limitations and potential areas where daily insulin may still have a favor specially in fine-tuning dosing. 

adjustments

Also, I have another concern regarding the economic aspects of once-weekly insulin compared with the once daily insulin. Is it cost-effective ?

Reply 2. Thanks for your comments and suggestions. Possible drawbacks of Icodec were briefly discussed. The leading concerns can be related to the fact that a long half-life of basal insulin may predispose to some problems in managing glucose variability. Other concerns may not exist in terms of a reduction in the frequency of glucose monitoring and the number of titrations compared to once-daily insulins. In fact, glucose monitoring can be carried out daily for both basal insulins to confirm the achievement of fasting/pre-breakfast glucose targets (e.g., 80 - 130 mg/dL). At the same time, insulin adjustment with once-daily basal insulins should be carried out every three days. However, it can also be advised to adjust the insulin dose once a week, as the same for insulin Icodec, especially for second-generation basal analogues (i.e., insulin Degludec U100 and U200 and insulin Glargine U300). Ref: https://diabeteseducatorscalgary.ca/medications/insulin/insulin-adjustments.html.

Costs of basal insulin largely depend on the mean insulin dose (estimated per day or week). On the other hand, costs depend on the organization of the health care system (private or public). The Italian health care system is public and has a regional distribution. To save public costs, novel and expensive drugs usually undergo prescriptive constraints by strictly selecting potential candidates, and drug costs are defined after region-to-factory negotiation to ensure 100% reimbursement of a specific medication. What is known so far is that Icodec costs can range from 1,100 to 1,500 dollars per year (around 1,300 assuming a mean daily insulin dose of 50 IU, https://www.cadth.ca/sites/default/files/DRR/2024/SR0790REC-Awiqli.pdf). The direct price is three times higher compared to insulin Degludec U100 or Glargine U300 (around 350 dollars per year). Specific large-scale, cost-effective trials are needed to comprehensively estimate the direct and indirect costs of Icodec compared to other once-daily insulin analogs. Please find changes in the text (changes are in red).

------

Question 3. 

Methods section: 

1. Why did the authors choose 24 weeks as a minimum duration for the included RCT? 

2. It is recommended to add PRISMA figure to illustrate the process of selection and the numbers of articles found in each step. 

3. There is no mention about who performed data extraction. Was it independently by multiple reviewers? 

4. The authors did not mention how did they handle missing data. 

Reply 3.

  • 24 weeks, around six months, were considered the minimum time to estimate the efficacy and safety of Icodec compared to once-daily insulin analogues
  • The PRISMA flow diagram was included (supplementary material, Figure S1)
  • Data extraction was carried out by two authors. Please check the text for details (changes are in red).
  • Missing data: Generally, the studies were well designed, and outcomes well reported. Missing data, such as lack of standard deviation or other dispersion measures, were handled as described in section 3.2.3.8. Software for statistics (in red).

-----

Question 4. Discussion section is written well however, the conclusion should be restructured in cohesive manner.

Reply 4. Thanks for your suggestion. The conclusion was re-written to cope with suggestions.

Reviewer 2 Report

Comments and Suggestions for Authors

Overall, the article provides a comprehensive review of Icodec in diabetes mellitus, but some minor revisions are needed before being accepted for publication.

1. Is it possible to retrieve all publications by searching only the PubMed/MEDLINE and Cochrane Library databases? Why haven't other databases, like Web of Science and Embase, been considered for retrieval?

2. The meaning of the symbols in Figure 2 needs to be annotated.

3. It is suggested to introduce the latest data on incidence rate of diabetes mellitus in the background.

4.The overall limitations of all studies included in the meta-analysis need to be stated in the discussion.

5. The conclusion should be made simpler. It is a condensed version of the results and discussions rather than a straightforward repetition of the findings.

Comments on the Quality of English Language

Minor editing of English language required

Author Response

Thanks to the reviewer for comments and suggestions. Here is a point-by-point reply to them.

Question 1. Is it possible to retrieve all publications by searching only the PubMed/MEDLINE and Cochrane Library databases? Why haven't other databases, like Web of Science and Embase, been considered for retrieval?

Reply 1. The review focused on the ONWARDS program to summarize the evidence from the updated randomized clinical trials currently available. However, we agree with your suggestion. So, according to the general recommendation to include at least three libraries, including databases and registry, we run an additional search on Clinicaltrial.gov. Please consider that the process of identification, screening, and inclusion of RCTs is illustrated in supplementary material Figure S1 (PRISMA 2020 flow diagram).

----

Question 2. The meaning of the symbols in Figure 2 needs to be annotated.

Reply 2. Thanks for your suggestion. Figure 2 was adjusted accordingly.

-----

Question 3. It is suggested to introduce the latest data on incidence rate of diabetes mellitus in the background.

Reply 3. Epidemiological data are included in the introduction (changes are in red).

-----

Question 4. The overall limitations of all studies included in the meta-analysis need to be stated in the discussion.

Reply 4. The limitations of studies are included in session 5.

-----

Question 5. The conclusion should be made simpler. It is a condensed version of the results and discussions rather than a straightforward repetition of the findings.

Reply 5. The conclusion is re-written to cope with your suggestions. 

Reviewer 3 Report

Comments and Suggestions for Authors

The review “Once-Weekly Insulin Icodec In Diabetes Mellitus: a Systematic Review And Meta-Analysis of Randomized Clinical Trials (ONWARDS Clinical Program)” is interesting and covers important aspects of the topic; however, I suggest a major revision before it can be considered for publication.

-          The abstract is concise and provides a good overview of the study's purpose, methodology, and key findings. However, consider expanding the results section to include specific data points and statistical outcomes that highlight the effectiveness of once-weekly insulin Icodec.

-          The introduction provides a solid foundation on the significance of diabetes management and the potential benefits of once-weekly insulin Icodec. However, a more detailed discussion on the current limitations of daily insulin therapy and how once-weekly insulin addresses these could strengthen the rationale for the review.

-          The resolution of figures should be improved.

-          Addition concluding remarks and future direction should be added in the conclusion section.

-          The aims of this review should be detailed in the last paragraph of introduction.

-          In the title Searching, screening, and selection of studies, Kindly update the inclusion and exclusion criteria in a separate title.

-          Citations: Ensure that all references are up-to-date and relevant. Consider including any recent publications that may have emerged during the review process.

-          

Comments on the Quality of English Language

Language and Style: The article is generally well-written, but a final review for grammar, style, and consistency in terminology would be beneficial.

Author Response

Thanks to the reviewer for comments and suggestions. Here, you can find a point-by-point reply to them.

Question 1. The abstract is concise and provides a good overview of the study's purpose, methodology, and key findings. However, consider expanding the results section to include specific data points and statistical outcomes that highlight the effectiveness of once-weekly insulin Icodec.

Reply 1. The abstract has been adjusted accordingly (changes are in red).

----

Question 2. The introduction provides a solid foundation on the significance of diabetes management and the potential benefits of once-weekly insulin Icodec. However, a more detailed discussion on the current limitations of daily insulin therapy and how once-weekly insulin addresses these could strengthen the rationale for the review.

Reply 2. The introduction now includes some drawbacks of Insulin Icodec, introducing the aim of the systematic review (changes are in red). 

-----

Question 3. The resolution of figures should be improved.

Reply 3. The resolution of the figures is 300 dpi, as requested by the journal policy. 

----

Question 4. Addition concluding remarks and future direction should be added in the conclusion section.

Reply 4. Please consider that the conclusion was re-written to accommodate the reviewers' request.

-----

Question 5. The aims of this review should be detailed in the last paragraph of introduction.

Reply 5. Thanks for your suggestion. The aim of the systematic review was included in section 2 after explaining the benefits and drawbacks of insulin Icodec (changes are in red). 

----

Question 6. In the title Searching, screening, and selection of studies, Kindly update the inclusion and exclusion criteria in a separate title.

Reply 6. The inclusion and exclusion criteria of trials are split into two separate sections, 3.2.3.2. and 3.2.3.3 (changes are in red).

-----

Question 7. Citations: Ensure that all references are up-to-date and relevant. Consider including any recent publications that may have emerged during the review process.

Reply 7. After the peer-review process, a careful re-check of records was made to follow PRISMA 2020 guidelines for the identification, screening, and selection of RCTs for the systematic review (up to 9 August 2024). No additional RCTs were found. One recent systematic review on the risk of hypos in T2D was identified and included among the references (n. 50).

Round 2

Reviewer 1 Report

Comments and Suggestions for Authors

The authors have addressed the comments and adjusted the manuscript accordingly. 

Reviewer 3 Report

Comments and Suggestions for Authors

The authors have improved its manuscript according to the reviewer's comments and suggestions. Therefore, I recommend the acceptance of this paper